# Learning Dynamics Models for Model Predictive Agents

## Abstract

Model-Based Reinforcement Learning involves learning a *dynamics model* from data, and then using this model to optimise behaviour, most often with an online *planner*. Much of the recent research along these lines (Chua et al., 2018; Janner et al., 2019; Yu et al., 2020) presents a particular set of design choices, involving problem definition, model learning and planning. Given the multiple contributions, it is difficult to evaluate the effects of each. This paper sets out to disambiguate the role of different design choices for learning dynamics models, by comparing their performance to planning with a ground-truth model – the simulator. First, we collect a rich dataset from the training sequence of a model-free agent on 5 domains of the DeepMind Control Suite. Second, we train feed-forward dynamics models in a supervised fashion, and evaluate planner performance while varying and analysing different model design choices, including ensembling, stochasticity, multi-step training and timestep size. Besides the quantitative analysis, we describe a set of qualitative findings, rules of thumb, and future research directions for planning with learned dynamics models. Videos of the results are available at https://sites.google.com/view/learning-better-models.

## 1   Introduction

Recently reinforcement learning (RL) (Sutton & Barto, 2018), in particular actor-critic approaches (Lillicrap et al., 2015) were shown to successfully solve a variety of continuous control problems (Schulman et al., 2017; Lillicrap et al., 2015; Fujimoto et al., 2018; Haarnoja et al., 2018). The simplicity of this approach has led to an explosion of research demonstrating the effectiveness of these methods (Vecerik et al., 2019; Kalashnikov et al., 2018). However, model-free RL suffers from two key disadvantages (Dulac-Arnold et al., 2021). First, model-free RL is sample inefficient, often requiring millions or billions of environment interactions. Second, the learned policies are tied to a specific task, making transfer of learned knowledge in multi-task settings or across related tasks difficult (Finn et al., 2017; Andrychowicz et al., 2017).

Model-Based RL holds the promise of overcoming these drawbacks while maintaining the benefits of model-free methods. Model-based RL involves learning a *dynamics model* from data, and then using this model to optimise behaviour, most often with an online *planner*. Since the dynamics model is ideally independent of the reward and state-action distribution, multi-task settings are simple: the reward function can be changed at will (Rajeswaran et al., 2020; Argenson & Dulac-Arnold, 2021; Belkhale et al., 2021). For the same reason, offline learning with off-policy data is possible (Yu et al., 2020; Kidambi et al., 2020; Argenson & Dulac-Arnold, 2021). Moreover, model-based RL can achieve a better sample efficiency (Janner et al., 2019; Nagabandi et al., 2018; 2020; Byravan et al., 2020; Buckman et al., 2018; Hafner et al., 2019b; Argenson & Dulac-Arnold, 2021).

In this paper we investigate the impact of the different model learning design choices on the planner performance. For this evaluation we focus on learning models using feed-forward neural networks as these are the dominant models in the recent model-based RL literature. A large portion of these papers mainly proposes different methods to train the networks and variations on the planner. Unfortunately, each new paper proposes a series of evolutions, and so individual design choices are rarely ablated fully. This is the goal of our work. We look specifically at the effects of various model learning design choices on the planning performance. We consider 4 different design choices, deterministic vs. stochastic models, multi-step vs. 1-step losses, use of ensembles, and input noise. These

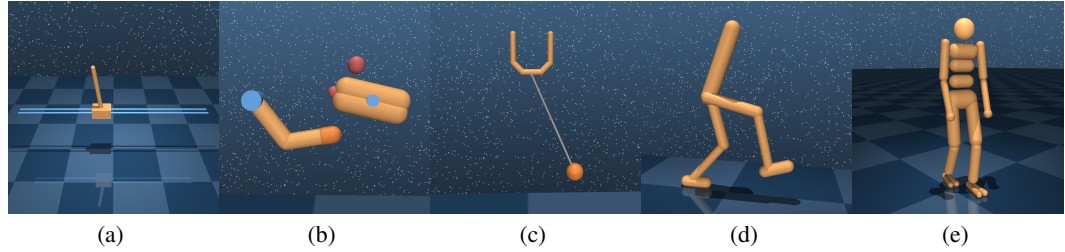

Figure 1: Domains of the DM Control Suite (Tassa et al., 2018) used in this paper. (a) cart-pole, (b) finger, (c) ball in cup, (d) walker, and (e) humanoid. These domains include various challenges like high-dimensionality, non-rigid strings and multiple contacts.

design choices have been proposed to obtain better long term prediction (Abbeel et al., 2005; Chua et al., 2018; Hafner et al., 2019b) and planning performance (Venkatraman et al., 2015; Sanchez-Gonzalez et al., 2020; Pfaff et al., 2020). To derive best practices for each choice-parameter, we perform ablation studies evaluating the impact of the different choices. In addition we investigate the qualitative performance of the learned models, by evaluating their ability to describe complex trajectories with reasonable fidelity, and to be consistent.

The contributions of this paper are

- We perform a systematic comparison evaluating the planning performance of different design choices for model learning, i.e., stochastic models, multi-step loss, network ensembles and input noise, on five DeepMind Control Suite environments.

- We observe that mean squared error (1-step or multi-step) is *not* a good predictor of planning performance when using a learned model.

- We find that multiple model learning design choices can obtain high reward and consistent long-term predictions for the evaluated systems excluding the humanoid. The differences in reward between the different combinations is not significant.

- We characterise best practices for learning models using feed-forward networks. For our experiments, deterministic models need to be trained using the multi-step loss and combined with ensembles. Stochastic models require more ensemble components and input noise.

The paper is structured as follows. First, we cover related work on model-based RL for continuous control in Sec 2. Afterwards, we review the naive approach to learning dynamics models (Sec. 3). Section 4 introduces the design choices of model learning and evaluates the learned models using model-predictive control. The subsequent discussion (Sec. 5) summarizes the insights, results and highlights general challenges of model learning for planning.

## 2 RELATED WORK

Learning dynamics model has a long tradition in robotics. The seminal work of Atkeson et al. (1986) proposed to learn the dynamics parameters of a rigid-body kinematic chain from data. To remove the limitations of analytic rigid-body models (Atkeson et al., 1986; Lutter et al., 2020; 2021), black-box function approximation was leveraged (Schaal et al., 2002; Choi et al., 2007; Calinon et al., 2010; Nguyen-Tuong et al., 2009; Nguyen-Tuong & Peters, 2010; Lutter et al., 2019; Lutter & Peters, 2019) to learn system models able to learn the peculiarities of real-world robotic systems.

The majority of model-based RL focuses on learning dynamics models, which predict the next step using the current state and action. The model learning for control literature refers to these models as forward models (Nguyen-Tuong & Peters, 2011). Model-based RL algorithms use the model as a simulator to generate additional data (Sutton, 1991; Janner et al., 2019; Morgan et al., 2021), to evaluate the reward of an action sequence (Chua et al., 2018; Nagabandi et al., 2018; Lambert et al., 2019; Nagabandi et al., 2020), to improve the value function estimate (Feinberg et al., 2018; Buckman et al., 2018) or backpropagate the policy gradients through time (Miller et al., 1995; Deisenroth

& Rasmussen, 2011; Heess et al., 2015; Byravan et al., 2020; Amos et al., 2021). Current model-based RL methods use 1-step predictions to compute the optimal trajectory. Most current model-based RL approaches use different architectures of deep networks to approximate the model, e.g., deterministic networks (Nagabandi et al., 2018; Byravan et al., 2020; Feinberg et al., 2018; Kuru-tach et al., 2018), stochastic networks (Heess et al., 2015; Lambert et al., 2019), recurrent networks (Hafner et al., 2019a;b; 2020; Ha & Schmidhuber, 2018), stochastic ensembles (Chua et al., 2018; Nagabandi et al., 2020; Janner et al., 2019; Kidambi et al., 2020; Yu et al., 2020; Buckman et al., 2018; Rajeswaran et al., 2020; Lambert et al., 2020b) and graph neural networks (Sanchez-Gonzalez et al., 2018).

In this work we focus on feed-forward neural networks for learning dynamics models with continuous states and actions. The learned models are used with model predictive control (MPC) (Garcia et al., 1989; Allgöwer & Zheng, 2012; Pinneri et al., 2020) to solve continuous control tasks. Simple feed-forward neural networks are the standard of most model-based RL and already offer many different variations that affect the planning performance. For the considered environments, MPC enables the direct evaluation of the learned models without requiring a policy or value function approximation. Therefore, the model performance for planning can be directly measured and compared to planning with the ground-truth model.

## 3 LEARNING DYNAMICS MODELS

We concentrate on 1-step forward models and review this setup in more detail. Dynamics models predict the next observation $\boldsymbol{x}_{i+1}$ using the current observation $\boldsymbol{x}_i$ and action $\boldsymbol{u}_i$. These models can be used as a simulator for planning without interacting with the real system. For example, some sample-based planning algorithms sample action sequences close to the current plan, evaluates the reward of each action sequence by simulating the trajectories using the dynamics model and updates the plan using the action sequences with the highest reward. The simplest approach to learn such a forward model is to use a deterministic model that minimizes the prediction error. This objective can be achieved by minimising the 1-step mean squared error between the predicted and observed next state. This optimization loss is described by

$$\theta^* = \arg\min_\theta \frac{1}{N_\mathcal{D}} \sum_{\boldsymbol{x},\boldsymbol{u} \in \mathcal{D}} (\boldsymbol{x}_{i+1} - \hat{\boldsymbol{x}}_{i+1})^T (\boldsymbol{x}_{i+1} - \hat{\boldsymbol{x}}_{i+1}) \quad \text{with} \quad \hat{\boldsymbol{x}}_{i+1} = f(\boldsymbol{x}_i, \boldsymbol{u}_i; \theta),$$

with the model parameters $\theta$, the dataset $\mathcal{D}$ with $N_\mathcal{D}$ samples.

**Dataset** Current model-based RL algorithms re-use the data collected by the agent to learn the model. Therefore, the data is acquired in the vicinity of the optimal policy. This exploration leads to a specialized model for the considered task as only regions of high reward are explored. Hence, the current learned models are not independent of the state-action distribution and the reward function cannot be interchanged at will. In contrast to model-based RL, system identification focuses purely on obtaining the optimal data for learning the model cover most of the state space.

**Integrator** Instead of predicting the next observation, one can predict the change of the observation. Then the system dynamics are modelled as an integrator described by $\hat{\boldsymbol{x}}_{i+1} = \boldsymbol{x}_i + \Delta t \, f(\boldsymbol{x}_i, \boldsymbol{u}_i; \theta)$ with the time step $\Delta t$. This technique has proven to result in more stable long term prediction compared to predicting the next state and has been widely adapted (Chua et al., 2018; Janner et al., 2019; Yu et al., 2020; Amos et al., 2021; Lambert et al., 2020a).

**Normalization** Optimizing the MSE can be problematic if the scales of the different observations are different. In this case, the model overfits to the dimensions with larger scales. For forward models this results in overfitting to the velocities as the position is usually an order of magnitude smaller. To mitigate this overfitting the MSE can be normalized using the diagonal variance of the dataset $\Sigma_\mathcal{D}$. The normalized MSE is described by

$$\theta^* = \arg\min_\theta \frac{1}{N_\mathcal{D}} \sum_{\boldsymbol{x},\boldsymbol{u} \in \mathcal{D}} (\boldsymbol{x}_{i+1} - \hat{\boldsymbol{x}}_{i+1})^T \Sigma_\mathcal{D}^{-1} (\boldsymbol{x}_{i+1} - \hat{\boldsymbol{x}}_{i+1}).$$

The normalized MSE is comparable to standardization of the model inputs and outputs. In our experiments we did not observe any significant benefits using standardization instead of the normalized MSE.

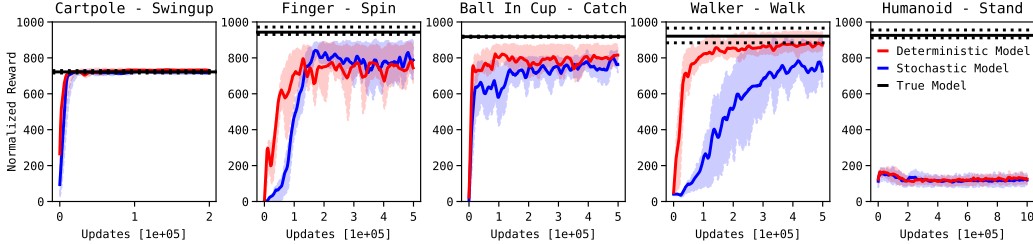

Figure 2: The learning curves comparing deterministic and stochastic models for five seeds on the five test domains. The shaded area marks the 20/80 percentile. For all systems except the humanoid, the stochastic and deterministic ensembles learn a good model that enables successful planning. The obtained reward is marginally below the reward obtained by planning with the true model. For the humanoid both approaches fail.

## 4 RESULTS

In the following we present our quantitative and qualitative evaluation of the model learning design choices. We evaluate the planning performance for 4 different design choices, i.e., stochastic models (Sec. 4.2), multi-step loss (Sec. 4.3), network ensembles (Sec. 4.4) and input noise (Sec. 4.5). These different design choices have been previously proposed within the literature to improve the model for planning (Abbeel et al., 2005; Chua et al., 2018; Hafner et al., 2019b) and generate better long-term predictions (Venkatraman et al., 2015; Sanchez-Gonzalez et al., 2020; Pfaff et al., 2020). For each of these proposed models we first introduce the approach, summarize the results and conclude the best practices. The best hyperparameter for model learning were identified first using random search and the ablation studies compare to the optimal hyperparameter. The humanoid is only used for the initial ablation study as no model-learning approach learns a model that could be used for planning. A more detailed experimental setup, the hyperparameters and additional ablation studies are contained in the appendix. The links within the results point to the referenced section contained in `https://sites.google.com/view/learning-better-models/`.

### 4.1 EXPERIMENTAL SETUP

**Data Sets** For the experiments the models are learned using a fixed data set. The data sets are recorded using five snapshots of an MPO policy (Abdolmaleki et al., 2018) during the training process. Therefore, the data set has diverse trajectories including random movements and optimal trajectories. The combined data set contains 500 episodes which corresponds approximately to 1 hour of data for the cartpole and 5 hours of data for the humanoid.

**Evaluation Criteria** The models are evaluated using the normalized task reward and the MSE. The normalized reward is rescaled such that the maximum reward per episode is 1000. The MSE is computed using the test set with an 80/20 split and normalized using the data set variance. If not specifically mentioned, the MSE is computed between a predicted and observed trajectory of 0.5s.

**Model Predictive Control** The planning performance of the models is evaluated using a CEM-based MPC agent (Garcia et al., 1989; Allgöwer & Zheng, 2012; Pinneri et al., 2020). Instead of maximizing the raw actions, we optimize the control points and interpolate linearly to obtain the actions at each simulation step. This approach simplifies the planning as the search space is decreased. We do not use a terminal value function. The MPC hyperparameters are optimized w.r.t. the true model and held fixed for planning with all learned models. For ensembles, we use a fixed ensemble component per rollout during forward prediction and do not switch between different components per rollout step. The reward is averaged across the rollouts of the ensemble (Chua et al., 2018; Lambert et al., 2019; Nagabandi et al., 2020; Argenson & Dulac-Arnold, 2021).

### 4.2 DETERMINISTIC VS. STOCHASTIC MODELS

Instead of using a deterministic model, Chua et al. (2018) proposed to use stochastic models. This model predicts the change of the system using a normal distribution. If the covariance matrix is

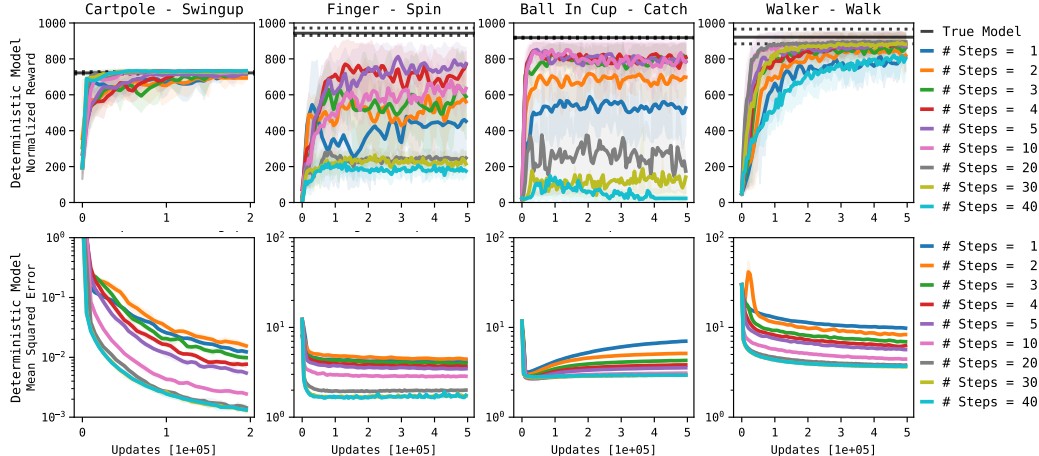

Figure 3: The learning and loss curves averaged across five seeds for different horizons of the multi-step loss. The MSE is computed between an observed and predicted trajectory of length $0.5$s. The shaded area visualizes the 20/80 percentile. For all systems, the multi-step loss for up to 5 steps improves the model performance for planning. For longer horizons, the MSE keeps decreasing but the obtained reward also decreases.

assumed to be diagonal, the next state is described by

$$\boldsymbol{x}_{t+1} = \boldsymbol{x}_t + \Delta t \left[ \hat{\boldsymbol{\mu}}(\boldsymbol{x}_t, \boldsymbol{u}_t; \boldsymbol{\theta}) + \hat{\boldsymbol{\sigma}}(\boldsymbol{x}_t, \boldsymbol{u}_t; \boldsymbol{\theta}) \, \boldsymbol{\xi} \right]$$

with $\boldsymbol{\xi} \sim \mathcal{N}(0, 1)$, the mean $\hat{\boldsymbol{\mu}}$ and standard deviation $\hat{\boldsymbol{\sigma}}$. Instead of minimizing the MSE, the stochastic model maximizes the 1-step log-likelihood described by

$$\boldsymbol{\theta}^* = \arg\max_{\boldsymbol{\theta}} -\frac{1}{2N_{\mathcal{D}}} \sum_{\boldsymbol{x}, \boldsymbol{u} \in \mathcal{D}} \left[ (\Delta\boldsymbol{x} - \hat{\boldsymbol{\mu}})^T \hat{\boldsymbol{\Sigma}}^{-1} (\Delta\boldsymbol{x} - \hat{\boldsymbol{\mu}}) + \log(|\hat{\boldsymbol{\Sigma}}|) - k\log(2\pi) \right]$$

with $\Delta\boldsymbol{x} = (\boldsymbol{x}_{i+1} - \boldsymbol{x}_i)/\Delta t$, the diagonal covariance matrix $\hat{\boldsymbol{\Sigma}}$ containing $\hat{\boldsymbol{\sigma}}^2$ and the state space dimensionality $k$. To make the optimization numerically stable the standard deviation is bounded by $[\boldsymbol{\sigma}_{\min}, \boldsymbol{\sigma}_{\max}]$. This clipping can be obtained using the differentiable transformation

$$\hat{\boldsymbol{\sigma}} = \boldsymbol{\sigma}_{\min} + S\left(\boldsymbol{\sigma}_{\max} - \boldsymbol{\sigma}_{\min} + S(\boldsymbol{\sigma}_{\max} - \hat{\boldsymbol{\sigma}})\right)$$

with the softplus function $S$. Depending on the implementation, the network can either predict the standard deviation (Hafner et al., 2019a) or the logarithm of the standard deviation (Chua et al., 2018). The common motivation for this stochastic model is the estimation of aleatoric uncertainty, which arises from the fundamental stochasticity of the environment. However, most performed experiments use a deterministic environment, which has no aleatoric uncertainty. The main benefits of the stochastic model for deterministic environments is the lower entropy bound and the adaptive selection of precision. The lower entropy bound reduces the risk of model exploitation by increasing the variance of the model predictions. The state-dependent variance enables a model with limited capacity to trade-off precision with higher variance at specific states. Prior work (Nagabandi et al., 2020; Hafner et al., 2019b) has also shown that fixing the variance to for all states can achieve the same performance as the state-dependent variance.

**Results** The learning curves comparing stochastic and deterministic models are shown in Figure 2. Videos of the model performance are available at [Link]. Both the deterministic and the stochastic ensemble learn a model that can be used for planning for most environments. For the cartpole, finger, ball in cup and walker the learned models are able to solve the task. The differences in reward are not large and the qualitative differences of the rewards are not visible. Compared to the true model, the obtained rewards are marginally lower and exhibit a higher variance. The differences between models are not clearly visible in the videos. Visualising the plans [Link] of the approximated models shows that the plans are plausible. Only for the ball in cup system the model exploitation is obvious as the ball can pass through the top of the cup. When transferring the models to a different task (Appendix - Table. 2), increasing the replan intervals (Appendix - Fig. 8) and comparing the learning

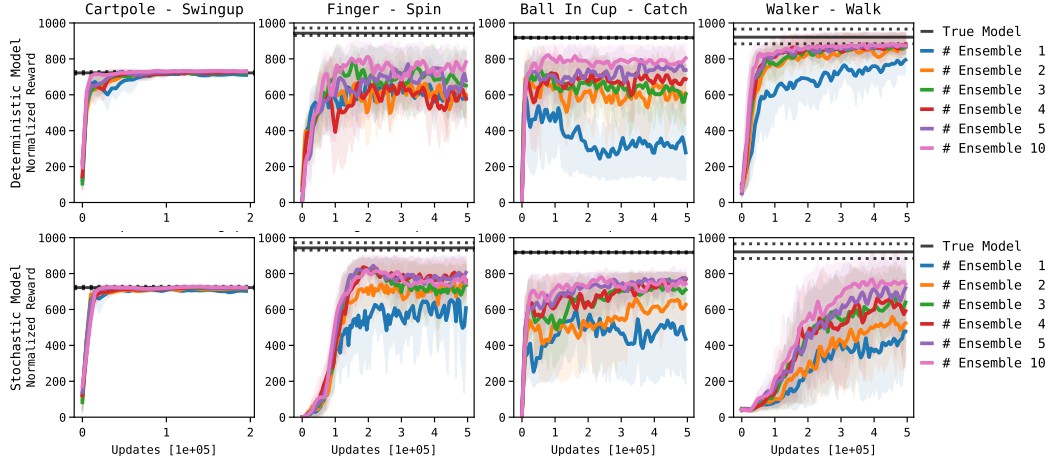

Figure 4: The learning curves averaged over five seeds for different number of ensembles. The shaded area highlights the 20/80 percentile. The obtained reward increases when increasing the number of ensembles. For stochastic ensembles the increase in performance is higher compared to deterministic ensembles. Without an ensemble both models do not fail at the task and only obtain a lower reward.

speed, the performance of the deterministic ensembles is better. For the humanoid both variants fail to learn a good model. The MPC agent is able to easily exploit the approximated model. The visualized plans show that the optimization finds an action sequence that lets the humanoid magically float midair [Link].

The results are partially different compared to Chua et al. (2018), which shows that the stochastic ensembles perform better than deterministic ensembles. We also found that stochastic ensembles are better, when the deterministic model was trained using the 1-step loss. However, if the deterministic model is trained using the multi-step loss (Sec. 4.3), the deterministic models achieve the same or better performance than the stochastic models trained on the 1-step log-likelihood.

**Conclusion** Both deterministic and stochastic ensembles are capable to learn models for planning for most evaluated systems and fail for the humanoid. However, both model types require different design choices to learn a good model for planning.

## 4.3   1-STEP LOSS VS. MULTI-STEP LOSS

The previously described losses optimized the 1-step loss that uses the information of two consecutive observations to learn the parameters. While this information is theoretically sufficient to learn the model accurately, various approaches (Abbeel et al., 2005; Venkatraman et al., 2015; Hafner et al., 2019b) have used the multi-step loss to learn better models. For the deterministic model the multi-step loss is described by

$$\boldsymbol{\theta}^* = \arg\min_{\boldsymbol{\theta}} \frac{1}{HN_{\mathcal{D}}} \sum_{\mathcal{D}} \sum_{i=0}^{H} (\boldsymbol{x}_{i+1} - \hat{\boldsymbol{x}}_{i+1})^T \Sigma_{\mathcal{D}}^{-1} (\boldsymbol{x}_{i+1} - \hat{\boldsymbol{x}}_{i+1})$$

with the horizon $H$, the prediction $\hat{\boldsymbol{x}}_{i+1} = \hat{\boldsymbol{x}}_i + \Delta t \, f(\hat{\boldsymbol{x}}_i, \boldsymbol{u}_i; \theta)$ and $\hat{\boldsymbol{x}}_0 = \boldsymbol{x}_0$. The computational complexity of the multi-step loss grows linearly with the horizon as the model needs to be evaluated $H$ times for each gradient update. Extending the multi-step loss to stochastic models is non-trivial. To compute the log-likelihood conditioned on the previous prediction one would need to propagate the uncertainty through the non-linear dynamics which is not possible analytically. Various approximations to propagate the uncertainty have been used for optimizing the policy (Deisenroth & Rasmussen, 2011; Watson et al., 2021), but these approaches have not been adapted to model learning, yet. Therefore, we do not consider these approximations within this comparison and leave it for future work.

**Results** The learning curves and loss curves on the test set are shown in Figure 3. The obtained reward for multi-step loss using up to 10 steps is higher than the reward of the 1-step optimization

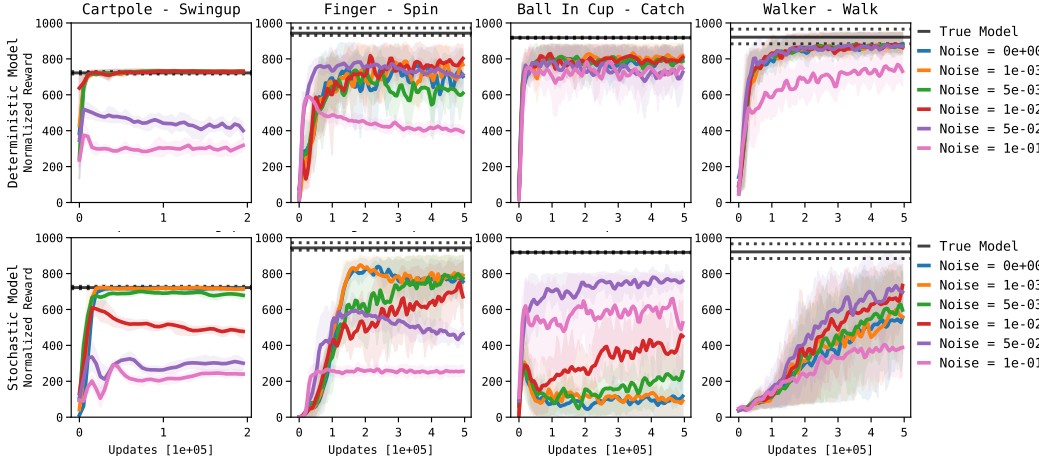

Figure 5: The learning curves averaged over 5 seeds for different levels of input noise. The shaded area highlights the 20/80 percentile. For deterministic models the input noise is not required to achieve a high reward. For stochastic models the input noise significantly improves the reward for walker and ball in cup. Without the noise ball in cup cannot be solved using the stochastic ensemble.

loss. For very long horizons the obtained reward drops for finger and ball in cup. The increase of the reward due to multi-step loss varies by system. For the walker and cartpole the increase is not large. For the finger and ball in a cup the multi-step loss is required to achieve a high reward. For all systems an horizon of 3 to 5 steps works best, which is much lower compared to the planning horizon of 50+ steps. In addition, we observed that increasing the horizon using a schedule results in a lower reward (Figure 9). The systems that benefit the most from the multi-step objective only obtain a reward comparable to the 1-step optimization loss. In contrast to the reward, the MSE keeps improving when increasing the horizon. This observation highlights that a lower MSE does not necessarily imply a better model for planning. We elaborate the implications of the MSE on the planning performance in section 5.

**Conclusion** For deterministic ensembles the multi-step objective increases the reward. Increasing the horizon using a schedule performs worse. A horizon between 3-5 steps performs the best for these systems. Despite the lower multi-step MSE for long horizons, the planning starts to fail for too long horizons. This observation highlights that the MSE (single or multi-step) is not a good indicator for the planning performance. A similar observation has been made for the 1-step log-likelihood by Lambert et. al. (Lambert et al., 2020a).

## 4.4 SINGLE NETWORK VS. NETWORK ENSEMBLE

To prevent the planner to exploit the model, various authors used the ensemble disagreement as a measure of epistemic uncertainty to regularize for the policy optimization. For example, PETS (Chua et al., 2018) averages over the reward of the predicted ensemble trajectories to prevent over-fitting to too optimistic predictions. MOReL (Kidambi et al., 2020) and MOPO (Yu et al., 2020) add a reward penalty to state with high epistemic uncertainty to prevent the policy optimization venturing into uncertain regions. The epistemic uncertainty can be approximated using bootstrapped network ensembles. The ensembles stack deep networks with different initialisation and train the networks with different minibatches. Therefore, all ensemble components are trained using the same data but the order and composition of the mini-batches is different. The computational and memory complexity for training the models and planning increases linearly with the number of ensemble components.

**Results** The learning curves for different ensemble sizes are shown in figure 4. Videos highlighting the model entropy [Link] and open-loop rollouts [Link] are available. More ensemble components increase the obtained reward. The increase in reward depends on the model type and specific system. For deterministic models the ensemble size only significantly matters for ball in a cup. For this task, the reward doubles when using a model ensemble. For the other systems the reward increases by up

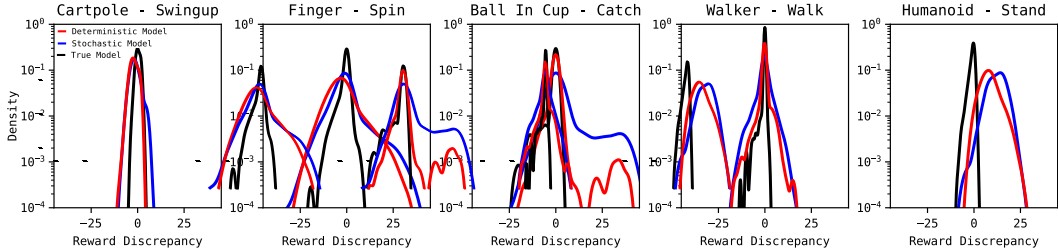

Figure 6: The reward discrepancy measuring the difference between the expected and observed reward. If the discrepancy is positive the planner is too optimistic. If the value is negative the planner is too pessimistic. The planner with the true model rarely over-estimates the future reward. The learned model frequently over-estimates the future reward highlighting the model exploitation. Especially for the humanoid the planner expects a too high reward in close to all time steps. For the other systems the model exploitation is not so drastic as the maximum mode ist at 0.

to 20%, which qualitatively does not significantly affect the task completion. For stochastic models, the advantages of ensembles is more significant compared to deterministic models. For all systems except the cartpole, the ensembles significantly outperform the single deep network. It is important to note that the computational complexity grows linearly with the number of ensembles. Therefore, future work needs to evaluate whether the increased performance of ensembles persists when the computational budget is fixed.

Qualitatively, the videos show that each ensemble component provides a stable long-term prediction of 1 second for all systems except the humanoid. The open-loop rollouts of the provide a diverse set of trajectories that are all physically plausible and not obviously wrong. However, the predictions are not perfect and have minor errors. For example, the walker marginally sinks into the floor and the edges of the cup are not modelled accurately. Only for the humanoid the predictions yield diverging trajectories that are obviously wrong. Furthermore, the entropy describing the epistemic uncertainty increases at intuitive state space configurations. For example the entropy increases when the finger touches the spinner or the walker is in contact with the floor.

**Conclusion** Adding more ensemble components to the model increases the planning performance. For the considered systems, the stochastic models benefit more using ensembles. The reward gain beyond five ensemble components is marginal. It is important to note that a single deep network does *not* completely fail at the task and only obtains a lower reward.

### 4.5 PERFECT OBSERVATIONS VS. INPUT NOISE

For dynamics models the data commonly lies on a manifold due to system constraints and over-parametrized observations. For example the cartpole positions with sine/cosine features lie on the cylinder. When predicting long time horizons using the approximate model, one will inevitably deviate from the manifold. This deviation is inevitable as even simulators with perfect assumptions and knowledge frequently violate constraints. Simulators use forces orthogonal to the constraint to return to the manifold. For learned models returning to the manifold is not straight forward as the dynamics are random outside the manifold. Therefore, it is very common that the policy optimization exploits these random dynamics during planning to obtain a high reward. Sanchez-Gonzalez et al. (2020) proposed to add noise to the inputs to learn a dynamics model that remains on the manifold for long horizons. Adding noise to the inputs causes them to lie outside the manifold. Using these inputs the model learns to predict solutions that return to the manifold. In this case the next step of the stochastic model is described by

$$\hat{x}_{t+1} = x_t + \Delta t \left[ \hat{\mu}(x_t + \lambda \, L_\mathcal{D} \, \omega, u_t; \theta) + \hat{\sigma}(x_t + \lambda \, L_\mathcal{D} \, \omega, u_t; \theta) \, \xi \right],$$

with $\omega \sim \mathcal{N}(0,1)$, the Cholesky decomposition $L_\mathcal{D}$ of the variance $\Sigma_\mathcal{D}$, and noise amplitude $\lambda$.

**Results** The learning curves for different noise amplitudes are shown in figure 5. Adding noise to the inputs can be beneficial, if the noise amplitude is not chosen too large for the specific environment. If the amplitude is too large the obtained reward drops significantly. For the deterministic models trained using the multi-step loss the input noise is not required to achieve a good performance. For

the stochastic models the input noise significantly improves the obtained reward for ball in cup and walker. Especially for ball in cup the adding inputs noise is required to obtain a high reward. Without the input noise, the stochastic model fails on this task.

**Conclusion** Adding input noise during the training can improve the model performance, if the noise amplitude is not too large. For some models this data augmentation is even required to solve the task, e.g., ball in cup with a stochastic model.

## 5 CONCLUSION

Feed-forward networks can learn models that enable planning for most of the considered systems. The obtained reward is only slightly lower than the reward when planning with the true model. Qualitatively, the models generate feasible plans and stable open-loop predictions for more than 100 steps. No single design choice works significantly better. Multiple combinations achieve a comparable reward. Deterministic models must be trained using the multi-step loss and combined with ensembles to achieve a good performance. Stochastic models trained using the 1-step log-likelihood perform well when ensembles are used and noise is added to the inputs.

**High Dimensional Systems** All tested models fail for the humanoid. The learned models predict diverging trajectories for open-loop sequences. The planner is too optimistic (Fig. 6) and finds an action sequence that lets the humanoid float midair [Link]. The results for the humanoid are expected as – to the best of our knowledge – only DreamerV2 (Hafner et al., 2020) obtained a humanoid model that is able to generate stable open-loop rollouts by using significant more data, online learning and a recurrent latent space model.

**Model Exploitation** The model exploitation is visible when comparing the difference between expected and obtained reward (Fig. 6). The approximate model frequently overestimates the reward, while the true model does not. Especially, systems that involve contacts of two moving bodies, e.g., finger and ball in cup, are more prone to exploitation. One common failure mode for ball in cup is that the planner tries to tunnel the ball through the bottom of the cup [Link]. The collisions of ball and cup are modelled for the open-loop trajectories. Only during planning, an action sequence is found that enables to bypass this collision. Therefore, the modeling the contact is not the main problem but the model exploitation in the vicinity of strong non-linear changes.

**Mean Squared Error** The experiments show that the MSE (multi-step or 1-step) is not a good indicator for planning performance. A similar observation was described by Lambert et al. (2020a) for the 1-step log-likelihood. It is important to note that this observation does not suggest that one desires a higher MSE model. The result only suggests that MSE is not sufficient to estimate the planning performance of the model. During training, the reward increases and the MSE decreases with the number of updates. However, the MSE does not provide information about the planning performance when comparing two trained models. A particular peculiar example of this effect is shown in figure 10. For ball in cup only a coarse time step obtains a high reward, even though the MSE is an order of magnitude higher compared to smaller time steps. It is intuitive that the accuracy is not that important for this task. The trajectory after the collision does not need to be modelled precisely. The model must only ensure that the ball can not tunnel into the cup.

**Future Work** To advance model learning for planning, the existing techniques must be (1) extended to high-dimensional observations and (2) made more robust to avoid model exploitation. As data fit is not the main indicator for planning performance and ensembles are not sufficient to prevent model exploitation, a promising research direction is adversarial model learning. This approach can explicitly incorporate the planner into the model learning objective to minimize the reward discrepancy and promises to learn a robust model. One should specifically focus on dynamics with strong non-linearities as these are more susceptible to model exploitation.

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

APPENDIX

## 6 EXPERIMENTAL SETUP

**Systems** To evaluate the model learning approaches, we apply these algorithms to Cartpole, Finger, Ball in Cup, Walker and Humanoid of the Deepmind Control suite (Figure 1). These systems cover a wide range of tasks, include contacts and include high-dimensional systems. For example the finger environment requires learning the interactions between the finger and the spinner. The ball in cup environment requires learning the collision shapes such that the ball does not tunnel into the cup. The walker and humanoid require to learn high dimensional dynamics and contacts with the floor.

**Data Sets** The data sets are recorded using five snapshots of an MPO policy during the training process. Therefore, the data set has diverse trajectories including random movements and optimal trajectories. The combined data set contains 500 episodes which corresponds approximately to 1 hour of data for the cartpole and 5 hours of data for the humanoid.

**Observations** The observations are over-parameterized and can include Cartesian coordinates or sensors. For example, the finger contains touch sensors on the tip. The humanoid observations contain the ego-centric positions of hands and feet. Therefore, the data does not correspond to minimal coordinates of the simulator. Due to the overparameterized representation, the data lies on a low-dimensional manifold defined by the kinematic structure. Therefore, not all points in $\mathbb{R}^n$ correspond to physically possible configuration. The system is fully observed. Hence, the simulator state - except the x-y world coordinates of the floating base systems - can be reconstructed from the observations.

**Reward Function** For environments with sparse rewards the reward function is smoothed to provide a dense function. Therefore, the reward corresponds to the euclidean distance to the goal state. This smoothing is necessary to provide a good search signal for the planner. The reward per step is bounded to be between 0 and 1. The reported normalized reward scales the reward such that the maximum obtainable reward for a single episode is 1000.

**Mean Squared Error** The reported MSE is computed using the test set with an 80/20 split and normalized using the data set variance. If not specifically mentioned the MSE is computed between a predicted and the observed trajectory of 0.5s.

**Model Predictive Control** The planning performance of the models is evaluated using a CEM-based MPC agent (Garcia et al., 1989; Allgöwer & Zheng, 2012; Pinneri et al., 2020). Instead of maximizing the raw actions, we optimize the control points and interpolate linearly to obtain the actions at each simulation step. This approach simplifies the planning as the search space is decreased. We do not use a terminal value function. The MPC hyperparameters are optimized w.r.t. the true model and held fixed for planning with all learned models. For ensembles, we use a fixed ensemble component per rollout during forward prediction and do not switch between different components per rollout step. The reward is averaged across the rollouts of the ensemble (Chua et al., 2018; Lambert et al., 2019; Nagabandi et al., 2020; Argenson & Dulac-Arnold, 2021).

## 7 ADDITIONAL EXPERIMENTS & ABLATION STUDIES

In addition to the experiments presented in the main paper, we provide additional experiments that vary the dataset (Sec. 7.1), the task for the trained model (Sec. 7.2) and the replan frequency (Sec. 7.3). We also provide ablation studies evaluating the impact of a schedule for the multi-step loss (Sec. 7.4) and the impact of the time discretization (Sec. 7.5).

### 7.1 MODEL GENERALIZATION W.R.T. DATASET

To test the performance across different datasets, we compare the planning performance when varying the dataset. We consider 5 different datasets that contain epsiodes with different average reward. For example *dataset* 0 contains only random actions while *dataset* 4 contains the converged MPO policy. The average and maximum reward of the datasets are shown in figure 7. The results show that the approach of learning a model and utilizing the model for planning achieves - in most cases - a higher reward than the average and maximum reward of the dataset. For the most domains the

Table 1: The hyperparameters for the deep networks and the model predictive control. The deep network parameters were identified using random search to obtain the parameters with the highest reward for each system. Ablation studies comparing different network parameters did not show any coherent trend for all systems. Therefore, these studies are omitted. The MPC parameters were optimized using random search using the ground-truth dynamics models.

**Deep Network Parameters**

| | Cartpole - Swing Up | | Finger - Spin | | Ball in Cup - Catch | | Walker - Walk | | Humanoid - Stand | |
|---|---|---|---|---|---|---|---|---|---|---|
| | Deterministic | Stochastic | Deterministic | Stochastic | Deterministic | Stochastic | Deterministic | Stochastic | Deterministic | Stochastic |
| Time Step [s] | 0.01 | 0.01 | 0.01 | 0.01 | 0.04 | 0.04 | 0.01 | 0.025 | 0.02 | 0.025 |
| Network Dimension | [2 x 128] | [3 x 256] | [4 x 512] | [2 x 512] | [2 x 512] | [3 x 512] | [3 x 256] | [3 x 512] | [3 x 512] | [4 x 512] |
| Activation | Swish | ReLu | Swish | Elu | ReLu | Swish | Swish | Elu | Elu | Elu |
| Learning Rate | 5e-4 | 1e-4 | 5e-4 | 5e-4 | 5e-4 | 5e-4 | 5e-4 | 1e-4 | 1e-4 | 1e-4 |
| Batch Size | 256 | 64 | 512 | 512 | 128 | 128 | 512 | 128 | 256 | 128 |
| Multi-step Horizon | 20 | - | 5 | - | 3 | - | 5 | - | 5 | - |
| # Ensemble | 5 | 5 | 5 | 5 | 5 | 5 | 5 | 5 | 5 | 5 |
| Input Noise | 0.0 | 1e-3 | 1e-2 | 1e-3 | 5e-3 | 5e-2 | 0.0 | 5e-2 | 1e-3 | 1e-3 |

**Model Predictive Control Parameters**

| | Cartpole - Swing Up | Finger - Spin | Ball in Cup - Catch | Walker - Walk | Humanoid - Stand |
|---|---|---|---|---|---|
| Horizon | 1.25s / 125# | 0.25s / 25# | 0.75s / 19# | 0.50s / 50# | 0.50s / 50# |
| Replan Interval [s] | 1/100s | 1/100s | 1/100s | 1/100s | 1/50s |
| # Control Points | 1/50s / 63# | 1/75s / 19# | 1/25s / 19# | 1/20s / 10# | 1/50s / 25# |
| # Particles | 200 | 500 | 200 | 200 | 500 |
| # Iterations | 1 | 1 | 1 | 1 | 1 |
| Exploration Noise $\sigma$ | 0.5 | 1.0 | 0.5 | 1.0 | 0.75 |
| Exploration Noise $\beta$ | 2.0 | 3.0 | 3.0 | 2.0 | 1.5 |
| $\arg\max_{\mathbf{u}}$ | True | True | True | True | True |

model-based planning approach obtains a high reward for *dataset 1* or *dataset 2*, which is far from containing the data of the optimal policy. Therefore, the learned models are capable to generalize the beyond the state-action distribution of the dataset. The generalization is not perfect as for the random dataset, the performance is commonly worse. However, this is to be expected as the random dataset often does not contain large portions of the state space. For example, the random actions dataset for the cartpole do not contain states close to the balancing of the cartpole. Therefore, the planner with the learned model can swingup the pendulum but not balance the pendulum. When comparing deterministic and stochastic models, the deterministic models achieve in most cases a better or identical performance than the stochastic model. The only exception is *dataset 2* for ball in a cup.

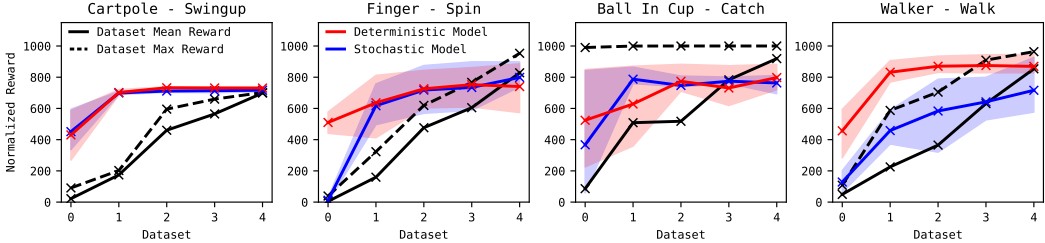

Figure 7: The obtained reward for the four different DM control suite environments and 5 different datasets with different average reward. In most cases, the model-based planning approach obtains a higher reward than the average dataset reward. Even when the dataset contains only random actions, i.e., *dataset 0*, the planner obtains a much higher reward than random actions.

## 7.2 MODEL GENERALIZATION W.R.T. TASK

To test the transferability of the learned models, we transfer the approximate models between the different walker tasks. For example, the model is trained using *walker walk* data and applied to solve *walker stand* and *walker run*. The results are summarized in table 2. The deterministic model performs well in the transfer tasks as the learned models generalize to the other two tasks. The obtained reward is marginally lower than training the model for the specific task. Only when transferring the deterministic model from *walker stand* to *walker run* the reward is much lower than training the model on *walker run*. This difference is expected as the state-action distribution between the stand and run task are very different. Furthermore, the experiments show that the transfer from the *run* to

Table 2: The obtained reward when transferring the model between tasks. The model is trained using the training domain and then applied to the evaluation domain. The deterministic model exhibits a good performance and obtains a comparable reward than the model trained on the test domain for most transfers. The stochastic model performs worse than the deterministic model when the model is transferred between tasks.

| Train Domain | Model Type | Walker Stand | | | Walker Walk | | | Walker Run | | |
|---|---|---|---|---|---|---|---|---|---|---|
| | | 20th Per | Avg | 80th Per | 20th Per | Avg | 80th Per | 20th Per | Avg | 80th Per |
| **Walker Stand** | Deterministic Ensemble | 820.1 | 897.8 | 976.4 | 787.8 | 853.5.5 | 949.5 | 344.4 | 394.6 | 446.2 |
| | Stochastic Ensemble | 468.4 | 690.0 | 905.5 | 296.6 | 426.0 | 565.6 | 87.0 | 140.9 | 190.9 |
| **Walker Walk** | Deterministic Ensemble | 890.0 | 931.3 | 993.2 | 822.9 | 869.0 | 962.3 | 463.5 | 503.6 | 563.6 |
| | Stochastic Ensemble | 685.4 | 782.9 | 886.3 | 672.8 | 750.2 | 930.6 | 212.5 | 300.7 | 411.5 |
| **Walker Run** | Deterministic Ensemble | 799.7 | 865.2 | 985.5 | 817.7 | 839.9 | 954.5 | 422.7 | 464.6 | 559.7 |
| | Stochastic Ensemble | 334.2 | 668.9 | 909.7 | 70.5 | 328.1 | 595.5 | 53.7 | 158.0 | 268.3 |
| **-** | True Model | 927.7 | 967.6 | 994.8 | 883.8 | 920.9 | 965.8 | 563.1 | 590.4 | 634.0 |

the *stand* task is simpler than transferring the model from stand to the run task. This difference is intuitive as the *run* datasets contains configurations of the standing walker, while the *stand* task does not contain the data of the fast moving walker.

When comparing the transfer of the deterministic and stochastic model, the deterministic model performs much better than the stochastic model. When transferred the 20th percentile of the deterministic model is better than the 80th percentile of the stochastic model for most task transfer. Therefore, the deterministic model is better when transferring models between tasks.

## 7.3 REPLAN FREQUENCY

An additional experiment to test the model performance for planning is to reduce the replanning frequency and observe the decrease in reward. As the replanning frequency is reduced, the planner evaluates fewer action sequences, is more likely to overfit to the approximate model and takes more steps before adapting the plan to the observed model mismatch. Therefore, the obtained reward can decrease for the true and approximate model. The results are shown in figure 8. For the cartpole and finger domain, the stochastic and deterministic model perform comparable. For the ball in cup and walker domain, the deterministic model performs better than the stochastic model. Especially for the walker domain, the deterministic model performance does not decrease while the reward of the stochastic model decreases by a lot. The obtained reward decreases slower / does not decrease for the deterministic model while the reward of the stochastic model decreases faster.

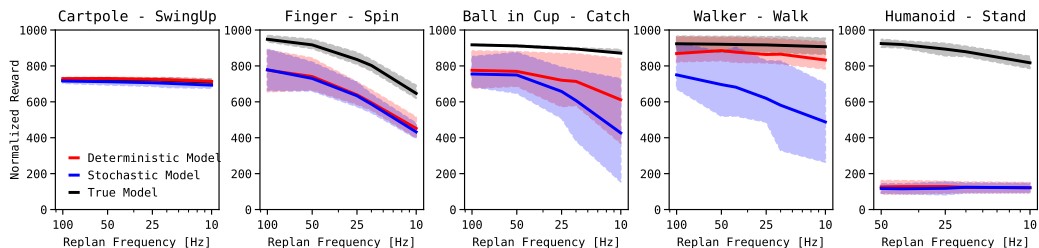

Figure 8: The obtained reward when decreasing the replan frequency for the deterministic, stochastic and true model. The shaded area highlights the 20/80 percentile. For the different replan frequencies, the deterministic model obtains a better or comparable reward compared to the stochastic model.

## 7.4 MULTI-STEP LOSS WITH SCHEDULE

Instead of training directly on the multi-step loss, one can increase the horizon of the multi-step loss using a schedule. One starts to train on the 1-step loss and increases the horizon with the number of updates. In this work we increase the horizon linear w.r.t. to the number of updates. The motivation for this schedule is that the multi-step loss is too ambiguous / hard to optimize this loss directly. Therefore, one uses a curriculum that starts with the simple 1-step loss and makes

the optimization slowly harder. The results for the multi-step loss using a schedule is shown in figure 9. In our experiments a schedule leads worse results than directly training on the multi-step loss directly. The MSE curves clearly show the effect of the horizon as the loss curves start identical. After some updates, the loss curves spread out according to the horizon length. Similar to training without schedule, the longer horizon lead to a better MSE. However, the MSE becomes unstable for some very long horizons. This instability was not observed without the schedule. Furthermore, the obtained rewards are comparable or worse compared to not using the schedule. This is especially visible for the *Finger spin* task. Even for the optimal horizon length of 5 steps, the schedule causes the model to obtain a reward comparable to 1-step loss, which is significantly worse compared to training without the schedule. Therefore, the model seems to over-fit to the 1-step and cannot obtain the same performance as training on the multi-step loss without schedule.

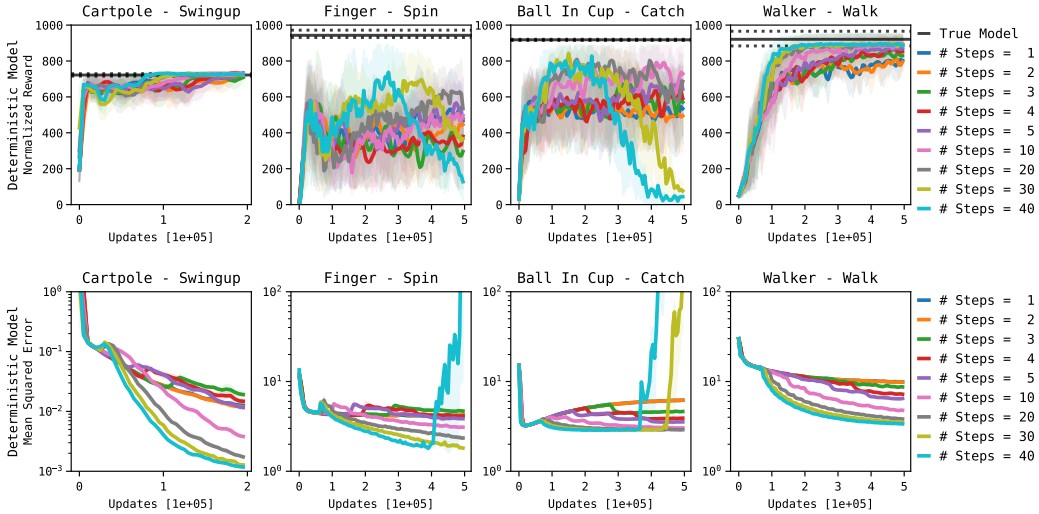

Figure 9: The learning curves and loss curves for the multi-step loss using a schedule to linearly increase the horizon. The schedule cause the loss curve to start identical and spread once the schedule increases the horizon. Compared to training on the multi-step loss without a schedule, the training with the schedule performs comparable or worse.

## 7.5 TIME DISCRETIZATION

A frequently neglected design choice for model learning is the choice of the time step. The time step can be chosen as a multiple of the observation frequency. The common choice of model-based RL approaches is to use the time step of the model-free RL agent, which uses control frequencies of 30 - 100Hz and up-scales the actions using action repeats. These control frequencies are much lower than the simulation time steps ranging from 100 - 500Hz for the considered environments. However, the time step can be chosen differently to simplify learning the model. The learning and loss curves for different time steps are shown in figure 10 and figure 11.

**Deterministic Model** If the time step is chosen too coarse, the MSE is much higher than a finer time step. Despite the lower accumulation error the MSE is higher. If the time step is too small, i.e., $< 1/100$s, the MSE starts increasing again due to the higher accumulation error. For most evaluated systems, the deterministic model obtained the smallest MSE for a model time step between $1/40 - 1/100$s. In terms of reward, the reward is lower for very coarse time steps as the time step lower bounds the control points distance for the planner. For example, the reward for *cartpole swingup* and *finger spin* the reward decreases for time steps greater than $1/100$s. A particular peculiar example is ball in a cup. For this domain, the planner only obtains a high reward when using a very coarse model with a time step of $1/20 - 1/25$s, despite that the MSE is about an order of magnitude larger than the MSE for a timestep of $1/50$s. This result shows that even the multi-step MSE is not a sufficient criteria to evaluate the model for planning. The optimal time step to obtain the highest reward depends on the domain. *Cartpole swingup* and *finger spin* require a fine time step of $1/100$s,

*ball in a cup catch* only solves the task using a very coarse time step of $1/25$s and *walker walk* performs well for time steps between $1/40 - 1/100$s.

**Stochastic Model** The optimal time steps for the stochastic models are identical to the optimal discretization of the deterministic model. Therefore, the optimal time step depends on the domain but not the model representation. In contrast to deterministic models, stochastic models perform slightly better in terms of reward for more coarse time steps. However, for very fine time steps the stochastic models perform worse than deterministic models. For example, the deterministic model solves the task for all time steps $< 1/10$s while the stochastic model fails for time steps $< 1/50$s. Furthermore, the MSE of *ball in a cup* shows that the stochastic models can diverge for very fine time steps.

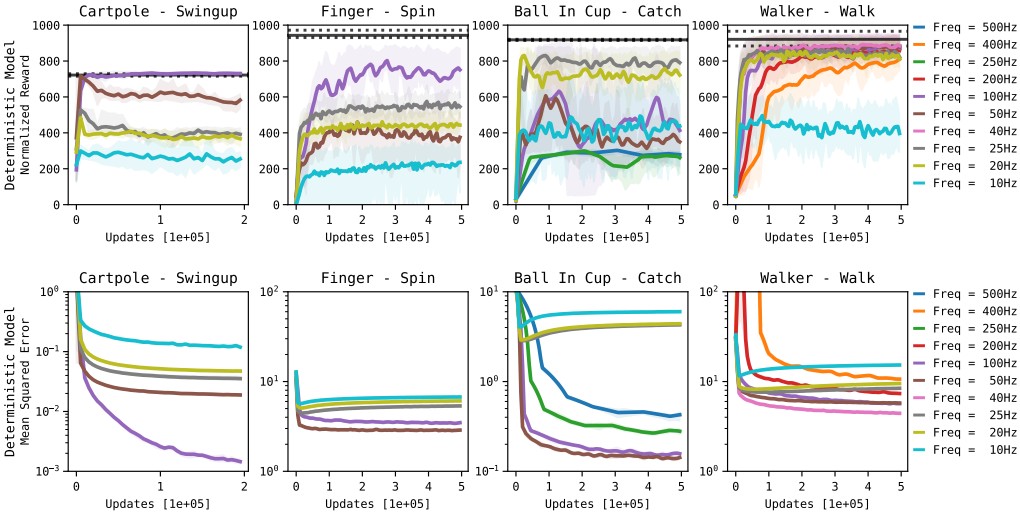

Figure 10: The learning and loss curves for different time discretization averaged over 5 seeds. The shaded regions highlight the 20/80 percentile. For too fine and too coarse time steps, the MSE increases. The lowest MSE is obtained for time steps between $1/40 - 1/100$s. The optimal time step to obtain the highest reward depends on the domain, while *cartpole swingup* and *finger spin* require a fine time step, *ball in a cup catch* only solves the task using a very coarse time step.

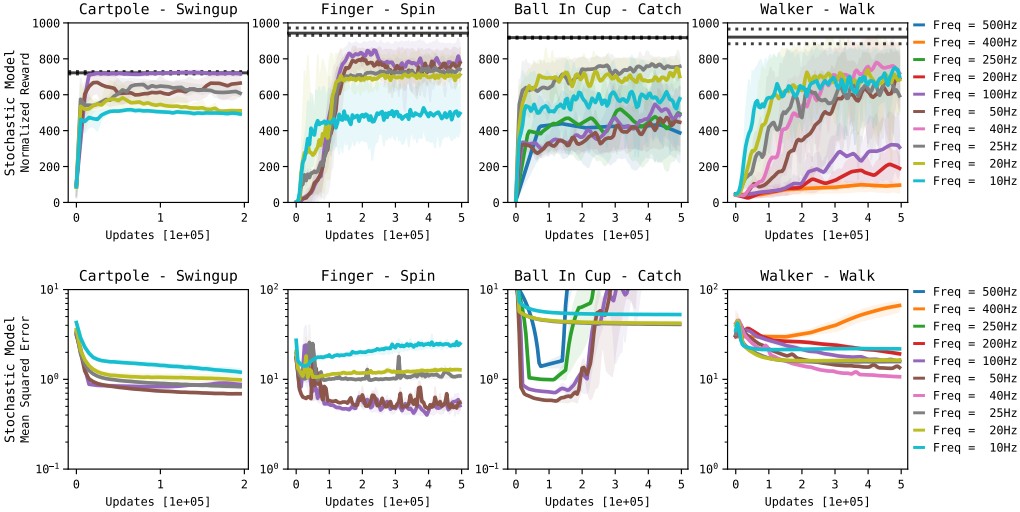

Figure 11: The learning and loss curves for the stochastic model with different time discretization averaged over 5 seeds. The shaded area highlights the 20/80 percentile. Similar to the deterministic model, the optimal time step depends on the domain. The optimal time steps are identical to the optimal time steps of the deterministic model.

