# OpenReview forum: "Learning Dynamics Models for Model Predictive Agents"
_ICLR.cc/2022/Conference — ICLR 2022 Submitted_

### Official Review · Reviewer_PwSp · 2021-10-28

**Correctness:** 4
**Technical Novelty And Significance:** 1
**Empirical Novelty And Significance:** 3
**Recommendation:** 6
**Confidence:** 4

**Main Review:**

Strong:
* I like the overall idea of the paper, not to propose another new method, but instead make a systematic comparison of some important aspects of model-based RL. I think we could use more papers like this.
* The paper is clearly written, has many results, and good additional videos online.

Weak:
* One of my doubts is the diversity of the tasks. Since the Humanoid does not give any results, they essentially compare performance on 4 tasks. Given that this is a benchmarking paper, I think this is a somewhat narrow range. I think the results partially illustrate this, since on multiple occasions the results are somewhat inconclusive (on one or two it works better, on the other two on parr or worse).
* I think you missed some important related work [1], which already extensively studied the effect of multi-step losses in RL tasks.
* I doubt whether it makes full sense to study stochastic models on deterministic environments only (Sec 4.1). I see that they may still be beneficial, but to really say something about the necessity of stochastic predictions, I think you would need to at least have a few stochastic environments as well.
* You decide to not propagate uncertainty over multiple steps because it cannot be done analytically through non-linear neural networks (Sec 4.3), which is true, but you can easily use a particle method for this right, i.e., use sampling? This is a common solution.
* While is the paper is generally well written, you incidentally have some strange sentences with missing words. For example: “Therefore, the modeling the contact is not the main problem but the model exploitation in the vicinity of strong non-linear changes”. I think the paper needs one more careful proofreading.

[1] Chiappa, Silvia, et al. "Recurrent environment simulators." arXiv preprint arXiv:1704.02254 (2017).

**Summary Of The Paper:**

This paper systematically studies a range of design considerations in model learning: 1) deterministic versus stochastic models, 2) 1-step versus multi-step targets, 3) the number of instances in an ensemble model, and 4) the effect of input noise to make the model more robust. They find 1) both deterministic an stochastic models work, 2) a medium horizon (3-5 steps) works best & model MSE is a poor indicator of planning performance, 3) ensembles improve performance up to about 5 components, and 4) the benefit of input noise varies over tasks.

**Summary Of The Review:**

This paper has positive and negative sides for me. As mentioned above, I like a comparison/benchmarking effort, which I believe the field indeed needs, and the authors have taken a nice, systematic approach, and written a well-structured paper. On the downside, I kind of doubt whether I learned too much from the results, since 1) the results vary quite a bit over tasks, which I think might be due to the relatively small set of tasks, 2) the results on stochasticity are limited by the set-up, which excludes stochastic environments, and 3) some other results were already partially known (like the multi-step prediction effects and poor performance of MSE). I am therefore a bit in doubt, because the paper does still have its merit in systematically comparing different element of model learning in deterministic control tasks.

---

> ### Author Response · Authors · 2021-11-22
> **Clarification Stochastic Multi-Step Loss**
>
> Yes, you could use the reparametrization trick to train the stochastic models on the n-step loss. However, the disadvantage of this approach is that the model has no incentive to retain the model entropy. While the 1-step log-likelihood balances accuracy and entropy the reparametrization trick solely focuses on accuracy. To avoid losing all the entropy one would need to add an entropy bonus to the stochastic multi-step loss. As this approach is not well established in the literature we did not include this stochastic multi-step loss within the paper. However, this approach is a promising direction for future work.
>
> Thanks for pointing out the missing reference. We will include this reference within the next update.

---

> > ### Comment · Reviewer_PwSp · 2021-11-23
> > **Response after author rebuttal**
> >
> > I read the rebuttal, and do agree with many things the authors write. A lot of computational effort has already gone into this paper, yet the results remain somewhat ambiguous, which is indeed frustrating. On the other hand, top conferences like ICLR should also raise awareness of the ambiguity of many of the methods we use in model-based RL. I would recommend the authors to slightly rephrase their claims ('ie., indicate that results are quite often ambiguous between methods'). I would want to raise my score to a 7, but the ICLR rating does not permit this, and I think an 8 is too much in its current form.

---

> > > ### Author Response · Authors · 2021-11-24
> > > **Author Response**
> > >
> > > Thanks for the positive response. I am happy to rephrase the claims in the final version and make it clearer that the results are ambiguous and don't let one select the optimal model learning technique for planning. Right now, I cannot update the paper draft anymore, but I will do it in the final version.

---

### Official Review · Reviewer_xfZq · 2021-11-02

**Correctness:** 3
**Technical Novelty And Significance:** 2
**Empirical Novelty And Significance:** 1
**Recommendation:** 3
**Confidence:** 3

**Main Review:**

Strengths:
* Interesting problem setting of understanding the role of design choices for model learning in model-based RL.

Weaknesses:
* Takeaways from the paper don’t seem to be conclusive (rf. “No single design choice works significantly better”). It doesn’t satisfactorily answer the question the work sets out. It might be helpful to frame the paper as informing future researchers who want to learn accurate dynamics models: How can this work help inform them which methods to try out vs. be eliminated?
* Investigation is limited in terms of environments explored. Specifically, the study only focuses on four control environments which are not very representative of settings where you would want to do model-based RL (e.g. the related works section motivates the use of accurate model prediction for robotics, but only applies it to simulated, deterministic environments). I would recommend the authors to extending their analysis in a robotics-based environment, e.g. RoboSuite [1], PlayRoom [2], Meta-World [3].
* Investigation is limited in terms of design choices explored. For example, memory-based architectures are omitted, despite playing a key role in domains that are partially observable (e.g. velocity of agents cannot be inferred from their setup which makes model-learning hard). It would be nice if their analysis could be expanded to cover memory-based architectures.

[1] robosuite: A Modular Simulation Framework and Benchmark for Robot Learning
[2] Learning Latent Plans from Play
[3] Meta-World: A Benchmark and Evaluation for Multi-Task and Meta Reinforcement Learning

**Summary Of The Paper:**

This work ablates some of the design choices that go into learning a dynamics model for control-based environments. They ablate 4 choices: use of deterministic vs. stochastic models, multistep losses, network ensembles, and input noise. The authors study this in a few of the DeepMind control suite environments.

**Summary Of The Review:**

--- methodology ---

I find the question of the paper interesting, but the methodology doesn’t feel satisfactory to me. A few things appear odd to me:
* Running a stochastic model in a deterministic setting seems like a weird setting for evaluating the benefits of stochastic models. Feels like an unfair comparison. Could the authors try running this on environments that elicit stochasticity, and where this choice might seem more natural?
* The setup of the paper seems odd: The data collection assumes that we already have a well-trained model-free agent that already does the task we care about...We then learn a dynamics model from this policy and deploy the model-based version. It seems that this setup doesn’t even need a model-based method in the first place. So why should we care about these domains that the authors evaluate on? Plus, the most challenging domain - humanoid - doesn’t seem to lend itself to model analysis in this work.

--- paper narrative & motivation ---

In some sense, the takeaway of the paper feels negative to me: there isn’t really a consistent way of learning accurate / useful dynamics models, so future researchers might just need to empirically test all the design choices for themselves and see what works best. I wish the work could do more to explore a bit more into what kinds of errors these dynamics models incur, and why these approximation errors are fatal - e.g. humanoid completely fails and so all of its results are omitted, but … why are those model errors that bad compared to other domains? How can this failure case inform us in how to better learn dynamics models?

There are several areas in the paper that come unmotivated or are not well contextualized. For example, I don’t really understand the dynamics manifold argument in Section 4.5 when the authors say that long time horizon predictions lead to inputs being outside the desired manifold, yet claim that adding noise also “causes the inputs to lie outside the manifold” but the model is more robust to predict solutions returning to the manifold. It seems like noise will only, up to a certain extent, help you in the short run but long horizon settings will still be difficult to predict accurately.

Why would we consider having a model-based method in the first place? It would have been nice to see explicit sample gains in having model-based methods - e.g. number of data samples saved for these settings [or considering some form of computational budget].

The work also seems to assume the best case scenario of having an optimal (model-free) policy which is oftentimes not available. It would have been interesting to investigate the approximation tradeoffs when a well-trained policy is not available as a data source for the to-be-learned dynamics model.

--- experiments ---

* What does the MSE graph look like for Humanoid?
* Observation that MSE is not a good predictor of planning performance when using a learned model seems non intuitive and potentially an interesting thing to further investigate.
E.g. Figure 3: Why is it that for longer horizons, MSE goes down and reward goes down? Since you only execute the first action anyway, how frequently does the plan deviate from what an optimal policy would on average do?
If you were to enforce the proper physical constraints in the prediction, does that resolve the exploitation problem? Since you have access to the simulator and you’re only predicting observation joints, this feels like a conjecture that can be easily verified. This would also disentangle whether physical constraints are the issue vs. the prediction being wrong at critical states of the trajectory.
* Why did the authors choose to use MPO policy as the base policy?
* Could the authors elaborate why the multi-step loss helps up to a certain number of steps for those domains?
* Is state dimensionality the core issue in preventing humanoid from working? Could this hypothesis be testing on 4k+1 observation-space swimmer environment where k varies?
* It’s impossible to have complete coverage of previous methods, but I feel like this work is missing a key class of dynamics models which are a function of history. E.g. in robotics settings (already cited by the authors), this includes concatenating states or having a recurrent neural network. This helps in settings where the environment is partially observable. Another thing is that it’s common practice to predict the difference in states. Authors should consider including experiments there.
* Related work is tied to robotics (and this is where a good dynamics model is super critical, not the synthetic simulated environments used here), but no robotics-like environment is used for evaluation.
* It’s surprising to see that ball-in-cup can (a) tolerate that much noise and (b) improves with increasing noise. Why is that the case for this domain, and not for the others? Eg. I could imagine Finger-Spin to also somewhat tolerate noise, but the fact that it’s sensitive makes me think that the observation ranges matter a lot in this study. Could the authors elaborate on this?

Nits:
* Title is not reflective of the paper goal. Currently reads quite broad - doesn’t indicate that this paper is about understanding the design choices for dynamics models.

---

> ### Comment · Reviewer_xfZq · 2021-11-23
> **Comment after reading rebuttal and other reviews**
>
> Firstly, I'd like to thank the authors for their detailed response. While I don’t disagree with the goal of the paper (a call to action to better understand model learning design choices), I don’t think the paper achieves what it describes in the abstract: “to disambiguate the role of different design choices for learning dynamics models”.
>
> Looking at the other reviews, it seems like the common consensus amongst reviewers is that this work is either inconclusive or inconsistent with prior work. I still think more scientific investigation is needed before publication. I say this not as discouragement towards these kinds of works from being submitted or published, rather from a belief in doing these great investigatory endeavors right.
>
> As I noted in my initial review (“It’s impossible to have complete coverage of previous methods”), I agree with the authors’ that “we need to make assumptions […] to a tractable sub-domain.” Nonetheless, speaking just on my own behalf, I made additional recommendations on exploring other factors because the results felt inconclusive. Possibly other factors could have opened interesting analysis/takeaways/explanations.

---

### Official Review · Reviewer_nmSn · 2021-11-02

**Correctness:** 2
**Technical Novelty And Significance:** 3
**Empirical Novelty And Significance:** 3
**Recommendation:** 5
**Confidence:** 4

**Main Review:**

The paper shows the following advantage:

(+) This work tries to provide the community a deeper understanding of the predictive planning model on the observation space, by comparing several commonly used model designing choices and providing a conclusion on the choices that generally work better. The comparison takes the network structure, loss function, and stochasticity of models into consideration.

In the meanwhile, there remain several concerns.

(-) The stochastic and deterministic models are both applied to deterministic environments, while the stochastic model may lose part of its advantage in this case. Considering the comparison between the stochastic and deterministic model is one of the important experiments in this work, it may be worth classifying the environments into 2 corresponding types, then looking into how stochastic and deterministic models perform in each of them. It can also verify whether the current conclusion still holds in stochastic environments.

(-) Another concern is about the random seeds. There are 5 random seeds in total. The average and confidence interval based on 5 random seeds may not be accurate enough, therefore it is hard to justify whether some conclusions are valid or not. For example, in Section 4.4 and Figure 4, the result indicates that network ensemble improves the performance more significantly in stochastic models, while in the deterministic model, ensemble only matters in Ball-in-cup. However, when reading the plots, both Finger and Walker have overlapped areas when indicating the 20/80 percentile. It is not clear to me if the above conclusion still holds when adding more random seeds. As a paper empirically compares different model designing choices, I believe showing reliable averaged performance is important. Thus, using more random seeds (ideally above 10) might make the conclusion more clear.

(-) It is mentioned in the paper that the experimental results obtained in this work are partially inconsistent with previous work (Chua et al. 2018), however, there is a lack of discussion on what may cause this inconsistency, which makes it hard to justify whether the result provided in this work is reliable or not.

(-) When comparing the stochastic and deterministic models, the networks have different structures. It causes differences including capacities and numbers of parameters to train. These differences can affect the learning efficiency of the agent. Thus it is not clear to me whether the conclusion about deterministic and stochastic models is accurate.


**Summary Of The Paper:**

This paper investigates different choices of designing the planning model which predict the next state on the observation space. The paper compares (1) prediction with deterministic or stochastic models, (2) using 1-step forward prediction loss or multi-step forward prediction loss to train the model, (3) prediction with single network or network ensemble, and (4) using perfect observation or add gaussian noise to the observation.
Through the above experiments, the paper suggests that deterministic models should be trained with multi-step loss, while stochastic models work better when the 1-step log-likelihood loss is used and noise is added to the observation space. In both cases, ensemble networks tend to show better performance.


**Summary Of The Review:**

Overall, I think the paper is not good enough to get accepted yet. On one hand, the paper focuses on understanding and comparing different settings of designing predictive models in the literature, which offers the community a better understanding of planning. On the other hand, there remain several concerns. My major concern is that the experiment setting may affect the accuracy of the conclusion provided in the paper. Focusing too much on deterministic environments makes me wonder if the conclusion still holds when there exists stochasticity in the environment. This may cause an unfair comparison between the deterministic model and stochastic model, which is a major part of this work. Furthermore, the overlapping area when showing the 20/80 percentile suggests more than 5 random seeds are needed. Moreover, using different network architectures can affect the learning efficiency thus causing inaccuracy when comparing these models.

---

> ### Comment · Reviewer_nmSn · 2021-11-23
> **Reply**
>
> I would like to thank the authors for their response. After reading other reviews and the authors’ responses, I tend to increase my score as the response answers parts of my questions&concerns, but I still think the paper is below the acceptance threshold.
>
> I totally agree that papers doing the systematic comparison are needed, as these papers provide us a better understanding of existing methods. However, considering that the current results showed no model consistently helps, I do think more clear guidance/conclusion is needed. For example, a careful classification of tasks or conditions might help with showing more clear conclusion. I also agree with reviewer xfZq that exploring other factors might help. Therefore, I think there still exist things to improve before the paper gets accepted, but I will also defer to the opinions of other reviewers.

---

### Official Review · Reviewer_sY5Q · 2021-11-03

**Correctness:** 3
**Technical Novelty And Significance:** 2
**Empirical Novelty And Significance:** 3
**Recommendation:** 6
**Confidence:** 3

**Main Review:**

I enjoyed reading this paper. The main weakness are that only four benchmarks were used to arrive at these conclusions, and some elements of the methodology are not clear.

I have some questions:
1) Data Used: If I understand sec 4.1 correctly, you are actually using fixed data sets to evaluate the rewards of MPC. While I understand you intentionally did this to ease comparison, you remove a key factor exploration of MBRL: exploration. This weakens your comparisons for stochastic models in particular, since the uncertainty of a model can interact with the planning algorithm to learn more efficiently (or not). I also do not really understand how this works, what are the "Updates" on the x-axis in figures, number of data points accessed from this fixed data set? I think it's important to be clear here, there is no connection with what the planner explores?

2) Are you drawing enough samples for the stochastic models? Stochastic models are very vulnerable to drawing too few samples. Some data on this would be useful.

3) Multi-step: Fig. 3 is curious, why are the results for deterministic, steps = 1 are worse (e.g. finger-spin, ball-in-cup) than those for deterministic 1-step prediction in Fig. 2? Are these 95% CI?

4) The multi-step results were the most surprising to me, the optimal number of steps also seems to vary considerably between tasks, from 5-40 steps. Any insight why this happens?

5) Ensembles in Fig. 4: Again, is not 1 ensemble the same as a regular NN in earlier figures? Finally, since 10 ensembles performs betst, what would happen if you used 15? Additionally, you basically get more samples (and uncertainty) by adding ensembles, so there could be a sampling factor impact results here as well.

6) Input Noise: Is this for the planner or the model learning? If you use a fixed data set, I don't see how this could be for the model? If it's for the planner, it seems more like state noise than *observation* noise (c.f. section title) since the future states will all be affected? Finally, shouldn't ball-in-cup noise=0 match earlier figures for stochastic model?

Minor:

- The figures are displaced 1-2 pages from the text, which is bit confusing.

- "The obtained reward is marginally below the reward obtained by planning with the true model." - The gap in mean reward seems to be >10% in most of these example, which is at least mathematically not marginal, even though their behavior may visually seem very close.

- Normalized MSE vs. standardization, did you try with state and actions that weren't centered around zero in your benchmarks? What you are doing here is basically the variance compensation of standardization, but not the mean one.

- Why in-text [Link] tags instead of /url clickable footnotes?

- It might be interesting to see if these model-based approaches actually could converge to near-optimal rewards if you let them run longers.

- Note: "Instead of using a deterministic model, Chua et al. (2018) proposed to use stochastic models. This
model predicts the change of the system using a normal distribution.". I would phrase it as "learns the variance of the predictions". MSE is of course mathematically identical to neg log-likelihood for Gaussian noise with isotropic covariance, it's just that the variance is never estimated (because parameters are invariant of this). What is new here is learning state-dependent noise, presumably modelled as an extra output of the network (often called aleatoric uncertainty as you note). I am not entirely sure what you mean by the lower entropy bound, perhaps just that the planner will tend to avoid poorly modelled parts of the state space since the noise is presumably higher?

- Typos: "rollouts of the provide", "significant more data", " the modeling the "


**Summary Of The Paper:**

The authors compare a number of recent and commonly used learning strategies for model-based planning on four typical RL control environments. The findings provides some useful insights for practitioners as well as for future research directions.

**Summary Of The Review:**

While some design choices made in the study could be clearer, and I would have preferred to have more benchmark environments, the paper is well written and the experiments appear carefully designed. I believe ablative studies like this are very useful for the research community and practitioners alike.

---

> ### Author Response · Authors · 2021-11-22
> **Clarification of Updates**
>
> The updates on the x-axis correspond to the number of gradient updates of the model parameters. As the data is fixed the x-axis does not correspond to the amount of data the model is able to see. It is more the number of minibatches the model has drawn from the fixed dataset to update the model parameters.

---

> > ### Comment · Reviewer_sY5Q · 2021-11-24
> > **Response to Rebuttal**
> >
> > I thank the authors for their response above and the general one. In general, I think this type of paper would be of value to the community. However, some important questions were seemingly not answered. In particular, I'm curious about the seeming internal inconsistencies in the benchmarks as outlined in questions 3 and 5. Shouldn't e.g. the 1-step deterministic model in Fig. 3 be the same as the deterministic model in Fig. 2? The confidence intervals on e.g. "Finger - Spin"  look very different. Am I missing something?

---

> > > ### Author Response · Authors · 2021-11-24
> > > **Author Response**
> > >
> > > I am happy to answer the questions in more detail.
> > >
> > > >Multi-step: Fig. 3 is curious, why are the results for deterministic, steps = 1 are worse (e.g. finger-spin, ball-in-cup) than those for deterministic 1-step prediction in Fig. 2? Are these 95% CI?
> > >
> > > The curves look different because Fig. 2 does not show the deterministic 1-step loss. Fig. 2 shows the optimal parameters for each model category. Therefore, the deterministic model in Fig. 2 is trained using the 5-step multi-step loss.
> > >
> > > > The multi-step results were the most surprising to me, the optimal number of steps also seems to vary considerably between tasks, from 5-40 steps. Any insight why this happens?
> > >
> > > While depending on the environment multiple multi-step parameters work well, 4-5 steps are usually the optimum in terms of reward and computation time. The reward does not increase by a lot when using more steps (most commonly it even decreases). However, the computation time increases linearly with the number of steps. Therefore, using 40 instead of 5 steps takes 8x as long as unrolling the model is sequential.
> > >
> > > We don't have any empirical evidence that explains the difference. It is also always hard to come up with experiments that give qualitative insights for non-trivial systems such as walker as the number of possibilities is so large.  From visual inspection, we could not identify clear causes. The coherent quantitative trend is that the reward gets worse when it is more than 10+ steps. The only exception is cartpole, whereas this environment is also by far the simplest as it does not include contacts.
> > >
> > > > Ensembles in Fig. 4: Again, is not 1 ensemble the same as a regular NN in earlier figures? Finally, since 10 ensembles perform best, what would happen if you used 15? Additionally, you basically get more samples (and uncertainty) by adding ensembles, so there could be a sampling factor impact results here as well.
> > >
> > > No, all previous figures showed an ensemble of 5 for each model. We always use the optimal hyperparameters when doing the ablation study, we think that this is the fairest approach, i.e., only varying one hyperparameter when fixing the other optimal parameters. The gain from 5 to 10 ensembles is not significant in terms of qualitative performance. Yes, it can increase the quantitative reward by <10% but increases the computation complexity and memory footprint by 2x. In this regime, one also comes close to max out the parallelization and GPU memory of a P100. Hence, we did not push it further as the reward, computation time & memory trade-off is not favorable.
> > >
> > > > Input Noise: Is this for the planner or the model learning? If you use a fixed data set, I don't see how this could be for the model? If it's for the planner, it seems more like state noise than observation noise (c.f. section title) since the future states will all be affected? Finally, shouldn't ball-in-cup noise=0 match earlier figures for the stochastic model?
> > >
> > > Again all previous models were trained using input noise as we always used the optimal hyperparameters. This noise is for the model learning as the input noise causes the training data to lie outside the data manifold. Therefore, the model is trained in the vicinity of the data manifold of physically plausible states. This approach can help for prediction. When predicting multiple timesteps one will inevitably diverge from the manifold of plausible states and having a trained model in this region stabilizes the models for long-term predictions. This approach was initially suggested by [1]. For them, it was essential to obtain stable long-term predictions.
> > >
> > > [1] Alvaro Sanchez-Gonzalez, Jonathan Godwin, Tobias Pfaff, Rex Ying, Jure Leskovec, and Peter Battaglia. Learning to simulate complex physics with graph networks. In International Conference on Machine Learning, 2020

---

> > > > ### Comment · Reviewer_sY5Q · 2021-11-26
> > > > **Reviewer Response**
> > > >
> > > > I thank the authors for their candid answers, which allays most of my concerns. The experiments seem thorough if somewhat limited (4 environments). If it took 1300 training runs to do hyperparameter optimization and ablate all model options, it's understandable that the scope has to be limited. Ideally, I would also have liked to see more analysis. However, I still think this paper contains valuable information as is. ICLR unfortunately does not allow me to raise my score to a 7 and I'm not sure I can defend an 8 given the limitations above.
> > > >
> > > > Minor:
> > > >
> > > > -Perhaps remind the reader of your ablation methodology when the seeming inconsistency in the results is first encountered in Fig.3.
> > > >
> > > > -The title could be more indicative of this being a benchmarking/ablation study of current methods, and not a new method.

---

> > > > > ### Author Response · Authors · 2021-11-29
> > > > > **Author Response**
> > > > >
> > > > > Thanks for the positive words. We will update the figure caption and make the title more indicative once, we can update the paper again.

---

### Author Response · Authors · 2021-11-22
**Rebuttal**

Dear reviewers, thanks for the reviews and sorry for the late rebuttal but I had to defend last week and did not have the bandwidth to focus on the rebuttal. We will address the concerns raised by multiple reviewers in this overarching rebuttal.

First of all, we agree with reviewer PwSp:
> “I like the overall idea of the paper, not to propose another new method, but instead make a systematic comparison of some important aspects of model-based RL. I think we could use more papers like this.”

We need more papers that evaluate/analyze certain aspects without trying to sell new ideas or showcase how good the proposed idea is. However, the current review experiences are unlikely to incentivize anyone to write these sorely needed papers. To write such papers, we need to make assumptions to narrow down the research question to a tractable sub-domain (in our case feed-forward networks & deterministic, fully observed continuous control environments). Analyzing the question, which model learning technique for model-based RL is the best in all environments is simply not doable within a single conference paper. Furthermore, we need to accept that these observed empirical results will be messy as there usually is not a single best method (in all of ML research. I have never seen one).

Within all our reviews we frequently read that reviewers wanted to have this assumption removed or this additional new experiment. For example, what about exploration, stochastic environments, partially observed environments, recurrent networks, etc. When analyzing one approach on a specific subdomain there will always be one more potentially interesting experiment that can be done. We among the authors have discussed these assumptions again and again. The assumptions that we used were our first step to narrow down what kind of model learning we want for model-based RL. It does not answer the grand question but it is a first step that nicely fits a conference paper.

#### **Conclusion & Take-Away Message:**
The paper concludes that there is not a single best model learning approach. When tuned correctly, both model categories can solve the tested continuous control problems except the humanoid. Therefore, we can only present hyperparameters that on average improve the performance. We do not interpret this as a negative result but understand it as a call to action! New model learning approaches are needed to get model-based RL working on the humanoid, which until now only Hafner et. al. have achieved without augmenting model-free data.

#### **General Approach of Ablation Studies (sY5Q, nmSn):**
Among the reviewers, there were some misconceptions about the hyperparameter configurations for the ablation studies. For the ablation studies, we first identified the optimal parameters for each model category & environment using random search (including network dimension, learning rate, activation, etc.). These parameters are summarized in Table 1. For each ablation study, we only varied the specific parameter and used the optimal parameters for the remaining parameters. In our opinion, this approach is the fairest as each model category can have different model architecture and hyperparameters. Therefore, we think that the ablation studies illustrate the difference between the modeling choices and are not affected by other hyperparameters. For example, Fig. 2 shows the optimal parameter for stochastic and deterministic ensemble components. Both models use an ensemble and the deterministic model is trained on the multi-step loss.

Only for the ensembles, we used 5 components instead of 10 components as the gains in reward are negligible but the computational complexity is 2x and significantly increases the runtime for all experiments.

#### **Stochastic Models for Deterministic Environments (nmSn, xfZq, PwSp):**
As the reviewers point out, using stochastic models for deterministic environments seems counterintuitive. However, since the publication of PETS (Chua et. al.), this model choice has been the standard practice. For example, MBPO (Janner et. al.), PDDM (Nagabandi et. al.), and many others use this class of models in deterministic environments. Therefore, our motivation was to test whether this standard practice is beneficial. Within the community, there are different intuitions about why these heteroskedastic models perform better. Some believe that these models can describe contacts better while others believe that they ensure a minimum entropy and smooth out model exploitation. While we don’t specifically consider these hypotheses, we do empirically show that deterministic models work just as well when they are trained on the multi-step loss instead of the single-step loss. (Regarding exploration and stochastic environments see next point.)

---

> ### Author Response · Authors · 2021-11-22
> **Part 2**
>
> #### **Assumptions  (sY5Q, nmSn, xfZq, PwSp):**
> We agree that there are specific assumptions made in the paper but these were made primarily to ensure that the ablation studies are feasible and to be able to cope with the combinatorial explosion. Therefore, we focused on fixed datasets, deterministic and fully observed environments. We agree that adding stochastic and partially observable environments are valid extensions but we believe that this is beyond the scope of the current study.
>
> We do not consider exploration, where stochastic models potentially benefit, as the main purpose of models is to generate accurate predictions and not help with exploration. When considering exploration one would also need to consider other model-based exploration techniques. It would be great future work to evaluate how different model choices affect exploration.
>
> We also do not consider recurrent models as we only use fully-observed environments. Therefore, the hidden state of the recurrent models is not needed to learn a perfect model. It would be great future work to look deeper into recurrent models for partially observed environments as these have obtained good results in the literature (e.g. Hafner et. al.).
>
> #### **More Seeds & Environments  (nmSn, xfZq):**
> Yes, more seeds and environments are always better. However, it remains unclear whether these additional experiments bring much more value. First of all, we only show ablation studies that have coherent trends and don’t make absolute statements about which model works better than the others. As these trends are coherent, we doubt that the take-away message will change with much more seeds. As the current results show that no model choice outperforms all the others, there will be no clear winner when adding additional environments. It would be great future work to look into different environments that require more abstract planning.
>
> As it is, the paper contains results from a total of 1310 training runs which take somewhere between 3 - 24 hours per run. In addition, we ran several training runs to find the optimal hyperparameters using random search and ran several ablation studies which did not make it into the paper as the results were not coherent. Given the amount of computational effort that has been used to generate the results for the paper, we do not see a benefit to running even more seeds as these are highly unlikely to change the paper’s conclusions.
>
> #### **Differences to Chua et. al. (sY5Q, nmSn):**
> We provide a discussion about the differences to Chua et. al. in the paper. When all models are trained using the 1-step loss, the deterministic models perform worse compared to the stochastic models trained on the 1-step log-likelihood (Compare Fig 2 & Fig 3 - 1 step). However, when comparing the 1-step stochastic models to deterministic models trained on the n-step loss, the deterministic models perform as well or slightly better than the stochastic model (Fig 2). Training the stochastic model on the n-step log-likelihood is not possible as this cannot be computed in closed form.

---

### Decision · Program_Chairs · 2022-01-20

**Decision:**

Reject

**Comment:**

The paper studies the effect of different design choices related to learning a dynamics model. The reviewers uniformly agree that the topic of the paper, systematically studying different design choices, is important. Furthermore, the paper is very well written. However, there are a number of weaknesses as well, that limit the relevance of this work. Arguably, the main weakness is that the results are inconclusive: there is no single design choice that is better, a conclusion that provides little guidance for researchers working in this space. Another weakness is that the study focuses on only 4 domains. And while performing such a study on a much broader set of domains can be prohibitively expensive, that doesn't take away from the fact that it is hard to draw strong conclusions from such a small set of tasks. For these reasons, I recommend rejection.